# Stapled Golgi cisternae remain in place as cargo passes through the stack

Gregory Lavieu*, Hong Zheng, James E Rothman*

Department of Cell Biology, Yale University School of Medicine, New Haven, United States

**Abstract** We have designed a membrane 'staple', which consists of membrane-anchored repeats of the trans-aggregating FM domain that face the lumen of the secretory pathway. In the presence of the disaggregating drug these proteins transit the secretory pathway. When the drug is removed these proteins form electron-dense plaques which we term staples. Unexpectedly, when initially positioned within the cis-Golgi, staples remained at the cis face of the Golgi even after many hours. By contrast, soluble FM-aggregates transited the Golgi. Staples and soluble aggregates placed in cis-Golgi cisternae therefore have different fates. Whereas the membrane staples are located in the flattened, stacked central regions of the cisternae, the soluble aggregates are in the dilated rims. This suggests that while the cisternae are static on the time scale of protein traffic, the dilated rims are mobile and progress in the cis → trans direction via a mechanism that we term 'Rim Progression'.

## Introduction

Anterograde transport (cis → trans) through the Golgi stack is a prerequisite for virtually all proteins that are ultimately secreted from the cell, targeted to the cell surface membrane, or localized in a plethora of internal membrane-bound compartments in plants and animals. Despite the fact that this fundamental process was first recognized over 50 years ago, there is still today no general agreement on how it works. The greatest distinction among the competing models for anterograde transport (reviewed in *Rothman, 2010*) is whether the 4–6 cisternae typically comprising the Golgi stack are proposed to be static or mobile over the time scale of protein flow across the Golgi (typically 5–20 min).

In 'static' models, a transport process such as budding/fusing COPI transport vesicles or possibly tubules is required to create anterograde flow of cargo.

In contrast, in 'mobile' models, the cisternae themselves continuously move as intact units in the cis → trans direction (termed 'cisternal progression'), forming at the cis face and being consumed at the trans face, so that no inter-cisternal transport process is required in the anterograde direction. In order to allow the Golgi stack to retain its resident proteins (such as glycosyltransferases) in the face of continuous cisternal turnover, mobile/cisternal progression models also require concomitant retrograde transport (trans → cis or Golgi → ER, or both) of steady-state Golgi resident enzymes, termed 'cisternal maturation' (*Glick and Malhotra, 1998*; *Glick and Luini, 2011*).

The strongest evidence for cisternal progression in Golgi stacks is that intrinsically large soluble cargoes such as immature collagen aggregates (*Bonfanti et al., 1998*)—which are far too large to be accommodated by COPI transport vesicles—can nonetheless be rapidly transported across the stack. On this basis it has been widely accepted by deductive reasoning, as distinct from direct visualization, that whole cisternae must indeed progress across the stack. This of course does not rule out that parallel processes of COPI vesicular transport (*Pelham and Rothman, 2000*) and/or tubule-based transport (*Trucco et al., 2004*) could also occur.

We sought to further test the cisternal progression paradigm by extending the deductive approach from soluble aggregates to aggregates fixed in the membrane, which should behave identically to

*For correspondence: gregory. lavieu@yale.edu (GL); james. rothman@yale.edu (JER)

**Competing interests:** The authors declare that no competing interests exist.

**Reviewing editor**: Suzanne Pfeffer, Stanford University, United States

**eLife digest** Most plant and animal cells contain an organelle known as the Golgi apparatus, which consists of a series of four to six stacked cisternae. Almost all the proteins that are secreted from the cell, or targeted to its plasma membrane, transit through the Golgi. This process takes roughly 5–20 min.

Although transport of proteins through the Golgi was first observed more than 50 years ago, it is still unclear exactly how this process occurs. One possibility is that proteins to be packaged move through the cisternae enclosed in vesicles, as if on a conveyor belt. Alternatively, the proteins themselves may remain stationary while the Golgi cisternae move over them.

Now, Lavieu et al. provide evidence that the Golgi shows both mobile and static behaviour depending on the type and size of the cargo being processed. To distinguish between these two mechanisms, they created a new type of protein cargo—which they called a 'staple'—that became fixed to the walls on each side of the cisternae and could not, therefore, move freely through the Golgi. They compared the processing of this protein to that of a more typical soluble protein cargo, which could move freely through the Golgi stack.

Surprisingly, the Golgi processed these two types of cargo in very different ways. The staples remained embedded in the walls in the center of the cisternae, whereas the conventional soluble cargo was transported past the staples and collected at the edges of the cisternae, which are known as rims. These are wider than the center of the cisternae, and the staples are too narrow to span them. Lavieu et al. suggest that the Golgi cisternae can be divided into two functionally distinct domains: the centers of cisternae, which remain stationary, and the edges or rims, which can move.

In addition to increasing our understanding of how proteins are prepared for transport inside cells, this new mechanism reconciles seemingly conflicting data by revealing that the Golgi can be both mobile and static.

soluble aggregates according to cisternal progression models. For this purpose, we designed reversible membrane 'staples' loosely modeled on the adhesion protein cadherin, which can adhere cells homo-typically. Cadherins consist of tandem repeats of an adhesive unit that binds itself preferentially in trans (*Pokutta and Weis, 2007*). A similar unit, which additionally is controllable with soluble ligands that can enter cells, is the FKBP mutant termed FM, which according to its crystal packing also interacts in trans (*Rollins et al., 2000*), with the additional feature that this interaction is disrupted by ligands such as AP21998 even in living cells (*Rivera et al., 2000*). This suggests that tandem membrane-anchored repeats of FM domains should mimic cadherin in producing a defined structure (staple) that will reversibly adhere intracellular membranes when it faces the cytoplasm, or in the present application produce intra-lumenal adhesions within Golgi cisternae when the FM repeats are lumenally-oriented. To our surprise, we found that luminal staples and compositionally analogous soluble aggregates placed in cis-Golgi cisternae had entirely different fates. Whereas the soluble aggregates were rapidly transported, the membrane staples remained in place.

## Results

### Design and properties of lumenal membrane staples in the ER

As the basic unit of a lumenal membrane staple we designed a chimeric protein, termed $CD8_{lumenal}$, that contained a signal sequence, for its proper insertion and translocation across the ER membrane, fused to 1) a fluorescent protein (GFP or DsRed) to track the protein within the cells by confocal microscopy, 2) four repeats of a domain (FM4) designed to self-aggregate in trans in the absence of the disaggregating drug (AP21998) (*Rollins et al., 2000*), and 3) the CD8 protein, known to be a trans-membrane protein ultimately targeted to the plasma membrane (*Hennecke and Cosson, 1993*; *Lavieu et al., 2010*). This $CD8_{lumenal}$ protein should initially reside in the ER oriented with its FM4-containing aggregation domain facing the lumen, and so long as it remains non-aggregated (in the presence of AP21998) should be effectively transported to the cell surface. On the other hand, if AP21998 is removed while in the ER—or at any later stage of transport through the Golgi—then intra-lumenal

adhesions are expected to form as CD8$_{lumenal}$ aggregates homo-typically in trans, like a cadherin. These staples are expected to remain in the ER if they are planar and too large to be packaged into a small COPII vesicle. Various structural models suggest a planar adhesion plaque will form that closely links two sides of ER lamellae or tubules, with an intra-lumenal separation between the stapled membranes that from first principles could vary from 10 to 23 nm (*Figure 1A*), depending on the preferred arrangement of trans-aggregating FM4 domains. In particular, however, if repeat FM domains interact like cadherins (*Miyaguchi, 2000*; *Pokutta and Weis, 2007*) primarily via their most-membrane distant module (i.e., at their tips), then we would expect a membrane separation in the stapled state close to 25 nm (model 1 in *Figure 1A*).

When CD8$_{lumenal}$ is expressed in HeLa cells in the absence of the disaggregating drug AP21998, numerous, discrete electron-dense plaques are in fact readily observed by electron microscopy (*Figure 1B*, red square). The staples trigger lumenal constrictions within ER tubules (*Figure 1C*) whose separation averages 25 ± 5 nm, suggesting that model 1 in which only the distal FM domains are typically bonded prevails in the ER (*Figure 1A*) most closely approximates the structure of the adhesion plaque. Remarkably, when we used cytosolic staples that harbor the FM domains at the cytosolic face of the membranes (CD8$_{cytosolic}$), ER membranes were artificially stacked, confirming the trans–nature of the staples interaction (*Figure 1—figure supplement 1*). Although we exclusively visualized staples resulting from trans-interactions (two opposed membranes), we cannot rule out that cis-interactions (on the same membrane) may occurs when the physical constraints are favorable.

At the confocal level, the staples showed a reticular pattern typical of ER. Only when AP21998 is added to disaggregate the staples in the ER is CD8$_{lumenal}$ transported to the cell surface (*Figure 2A*). Importantly, when the staples are stuck in the ER they did not prevent the production or transport of co-expressed Golgi resident enzymes (ST-RFP), showing that ER staples are not toxic (*Figure 2B*).

We also employed biochemical tests of cell extracts to characterize the staples. When tested by SDS-PAGE, the disaggregating drug increased Triton-X100 solubility (without sonication procedure) of CD8$_{lumenal}$ in cell extracts by a factor 3.9 ± 0.9 (*Figure 2C*). Note that this relatively modest response of CD8$_{lumenal}$ to the disaggregating-drug (44% Triton X100-solubility after 4 hr treatment (*Figure 2C*) can be attributed to the transient transfection procedure. A large portion of the transfected cells showed a very high level of CD8$_{lumenal}$ over-expression, and did not respond well to the disaggregating drug. As a consequence CD8$_{lumenal}$, remained aggregated in the ER even during drug treatment. This also explains the apparent moderate efficiency of transport of disaggregated-CD8 when measured by bulk assay (*Figure 3B*). Disaggregated CD8$_{lumenal}$, expected to be O-glycosylated (*Nilsson et al., 1989*; *Lavieu et al., 2010*), appeared as an isoform with reduced SDS-PAGE motility (* *Figure 2C*). As expected, Jacalin, a lectin that binds D-Galactose residues, exclusively precipitates this isoform (*Figure 2D*). In addition, metabolic radiolabeling experiments showed that staples (~5.5 hr half-life) were slightly more stable that disaggregated CD8$_{lumenal}$ (~2.5 hr half-life) (*Figure 2—figure supplement 1*). Such stability is enough to guarantee the presence of the staples during numerous complete rounds of anterograde transport either from ER to Golgi or across the entire Golgi (around 20 rounds if we assume that one round of transport takes 15 min). Note that staples persist in the ER and in the Golgi (*Figure 2—figure supplement 1*) for the same time period (5–6 hr) before degradation by a cellular machinery that is not yet known.

Together, these results showed that ER-aggregated CD8$_{lumenal}$ forms staples that remain within the ER during their lifetime. However, this is a reversible state because adding the disaggregating drug dissolved a portion of the staples and allowed disaggregated CD8$_{lumenal}$ to behave as a proper antero-grade cargo that is glycosylated in the Golgi prior to being targeted to the cell surface membrane.

## Fate of membrane staples introduced into the cis-Golgi cisternae

Temperature blocks have allowed the selective interruption of the secretory pathway in mammalian cells. A classic example is the 20°C temperature block, which slows exit from the trans-most Golgi cisterna (TGN) more than it slows ER → Golgi transport or transport from cis → trans Golgi (*Matlin and Simons, 1983*). As a result secretory cargoes accumulate in the TGN, from which they can be synchro-nously released by raising the temperature to the physiological value of 37°C. Temperature blocks in the 15–18°C range slow export of secretory cargo from the ER, but slow anterograde transport in the Golgi stack even more, resulting in limited penetration of the Golgi stack with accumulation in the ER–Golgi intermediate compartment (ERGIC) and the cis-most Golgi cisterna or two (*Hobman et al., 1992*; *Balch et al., 1994*; *Volchuk et al., 2000*).

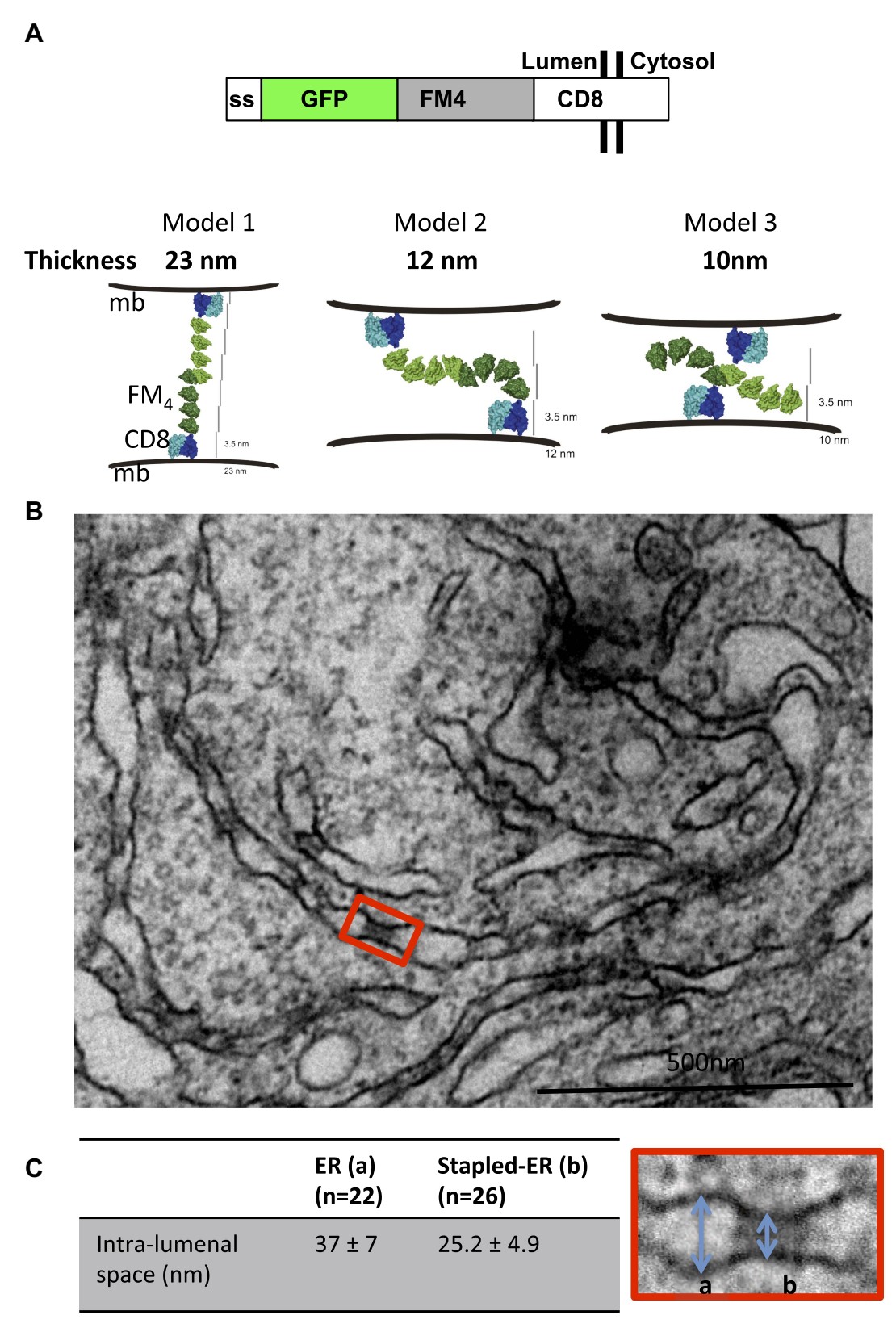

**Figure 1**. Aggregated-CD8$_{Lumenal}$ as an ER staple. (**A**) Domains harbored by CD8$_{lumenal}$. Predicted models of trans-interacting CD8$_{lumenal}$ preoteins that are expected to form lumenal staples. Predicted intra-lumenal space varies from 10 to 23 nm. (**B**) Electron-micrograph showing ER of HeLa cells expressing

*Figure 1. Continued on next page*

*Figure 1. Continued*

CD8$_{lumenal}$ in the absence of the disaggregating drug. ER-aggregated CD8$_{lumenal}$, which form ER-staples, appear as electron-dense flat features (red square). (**C**) Lumenal staples trigger lumenal constriction of ER tubules, consistent with model 1.

The following figure supplements are available for figure 1:

**Figure supplement 1**. Cytosolic staples stack the ER.

Following an established procedure (*Volchuk et al., 2000*) we employed a 16°C temperature block to load and accumulate some of the disaggregated CD8$_{lumenal}$ into the cis-Golgi of HeLa cells. Then, continuing at 16°C, we washout the disaggregating drug to allow for membrane staples reformation in the cis-Golgi. Finally, to explore the fate of these staples we raised the temperature to 37°C to allow anterograde transport to resume (*Figure 3A*). Unless it is explicitly stated to the contrary, all of these experiments were performed in presence of cycloheximide to prevent protein synthesis.

An independent, biochemical test confirmed that the unassembled CD8$_{lumenal}$ had reached the cis-Golgi but not the trans-Golgi at 16°C. Specifically, a portion of disaggregated CD8$_{lumenal}$ is precipitated by Helix Pomatia (*Figure 3—figure supplement 1*), a lectin that binds a N-Acetyl-Galactosamine residue that is only added within the cis-Golgi (*Roth et al., 1994*). In contrast, the lectin Jacalin, which requires trans-Golgi localized modifications for binding (*Spicer and Schulte, 1992*), did not precipitate CD8$_{lumenal}$ at 16°C (*Figure 3—figure supplement 1*), although it did at 37°C (*Figure 2D*).

To explore the fate of the staples when transport is permitted, we first used a surface biotinylation method and immunofluorescence in intact cells to assess the surface exposure of CD8$_{lumenal}$ after release of the temperature block to allow anterograde transport to resume. Surprisingly, the CD8$_{lumenal}$ re-aggregated in the Golgi at 16°C did not reach the surface and was retained in the Golgi as judged by confocal microscopy (*Figure 3B,C*, lane 2). Golgi retention of the staples is due to controlled aggregation, because this block to transport was removed when the drug was added back, and CD8$_{lumenal}$ (disaggregated) now moves from the Golgi to the plasma membrane (*Figure 3B,C*, lane 3). Note that except when mentioned otherwise, we used as a loading control 10% of the pellet fraction (insoluble fraction after sonication), which is almost free of soluble disaggregated-CD8$_{lumenal}$ (including the glycosylated-form). Again we attributed the apparent low efficiency of CD8$_{lumenal}$ transport to transient transfection. Cells that respond to the drug show >50% transport efficiency to the PM after 2 hr chase (*Figure 3C*, lane 3), consistently with the efficiency of transport of regular CD8-GFP (FM4-free) reported previously (*Lavieu et al., 2010*).

Next, we used confocal microscopy to further independently delineate where the staples accumulated within the Golgi at 16°C and their fate during a chase at 20°C which allows anterograde transport to proceed to the trans-Golgi but not beyond. We confirmed that co-expressed cis- and trans-Golgi fluorescent markers (Grasp65 and Golgin97, respectively) could be well discriminated by confocal microscopy and the degree of separation quantified with Pearson's coefficient (*Figure 4A* and graph). As a negative control we showed that ER-staples did not co-localize with either the cis- or trans-Golgi markers (*Figure 4B*, 4 and graph). Using the same method we demonstrated that the initial localization of membrane staples Golgi staples formed at 16°C overlaps with the cis- but not the trans-Golgi marker. This co-localization with the cis- but not the trans-Golgi marker persists following warm-up to 20°C (*Figure 4B*, 2, 3 and graph), implying that the staples do not migrate in the cis → trans direction within the stack, even under conditions where this transport process is expected to resume. Again, the block to transport of the staples is entirely due to aggregation, because adding the drug to disaggregate the staples now allows transport and increased co-localization with the trans-Golgi marker following the warm-up to 20°C (*Figure 4B*, 2 and graph).

Electron microscopy was used to further confirm the results from confocal microscopy. The staples were recognized in thin sections as electron dense plaques within Golgi cisternae (arrows, *Figure 5A–C*) spanning a gap of 27 ± 3 nm (*Figure 5D*) between opposing membrane surfaces, similar to their morphology within the ER. Staples formed at 16°C and then chased at 20°C for either 2 or 6 hr (which according to confocal microscopy remained in the cis-Golgi) were as would now be expected exclusively localized at one face of the Golgi, within the very first or more rarely the second and the third cisternae (*Figure 5A–C* and graph *Figure 7C* for quantification). Immuno-electron microscopy

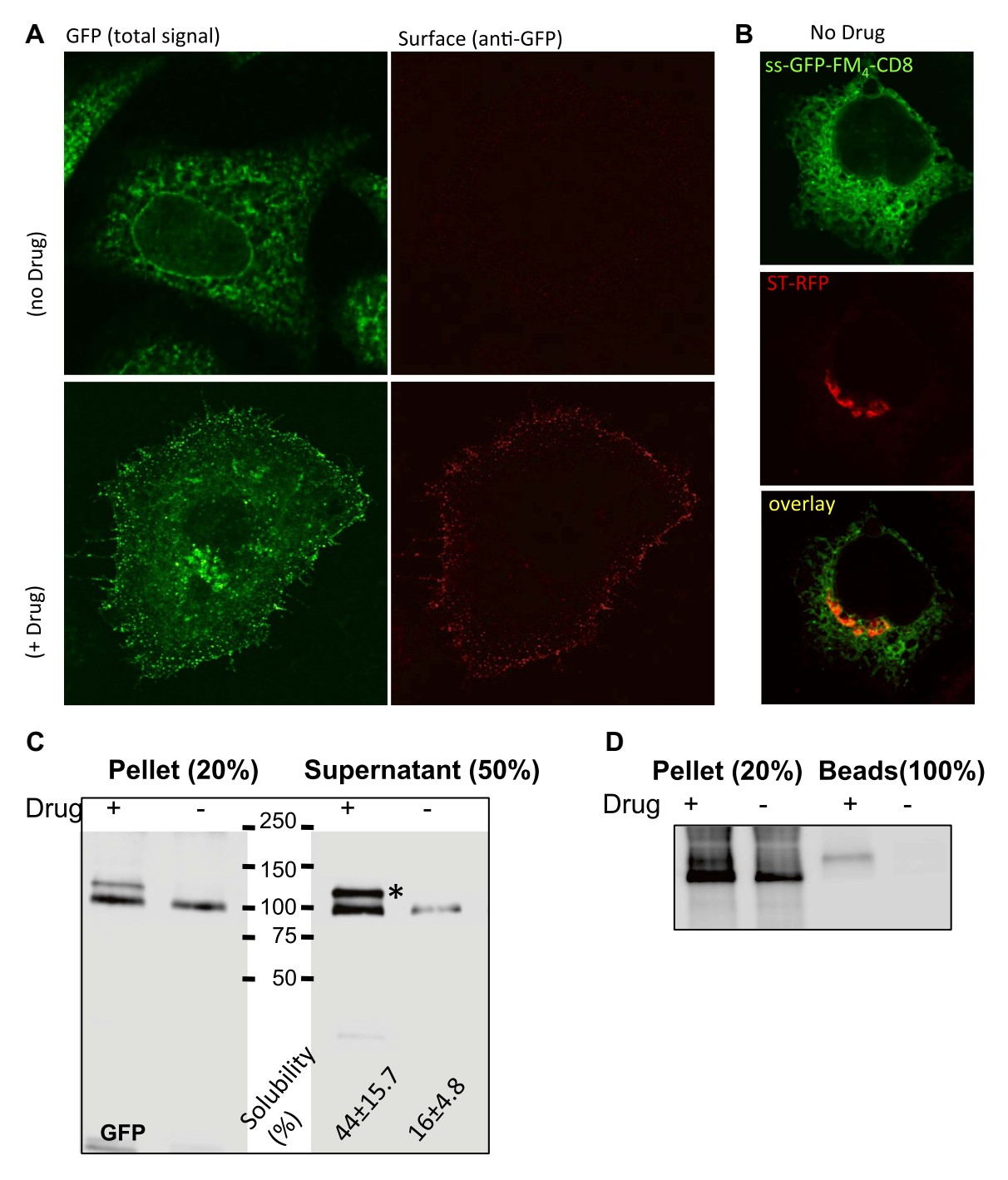

**Figure 2**. Disaggregated-CD8$_{lumenal}$ as an anterograde cargo. (**A**) Confocal micrograph showing HeLa cells expressing CD8$_{lumenal}$ in the presence or in the absence of the disaggregating drug. Without drug, ER-aggregated CD8$_{lumenal}$ remained at the ER (ER staples), whereas with drug (for 4 hr) disaggregated CD8$_{lumenal}$ is transported to the plasma membrane (PM), where it could be detected with an anti-GFP antibody in non-permeabilized cells. (**B**) Confocal micrograph showing HeLa cells co-expressing CD8$_{lumenal}$ with ST-RFP in the absence of the disaggregating drug. The ER staples do not alter Golgi targeting of ST-RFP. (**C**) Immunoblot showing that 4 hr treatment of HeLa cells with the disaggregating drug increases TritonX-100 solubility of CD8$_{lumenal}$. Disaggregated-CD8$_{lumenal}$ shows reduced mobility on SDS-PAGE gel (* upper band). (**D**) Immunoblot showing that Jacalin, a lectin that binds galactose residues, exclusively precipitates disaggregated CD8$_{lumenal}$.

The following figure supplements are available for figure 2:

**Figure supplement 1**. Lifetime of staples and disaggregated CD8$_{lumenal}$.

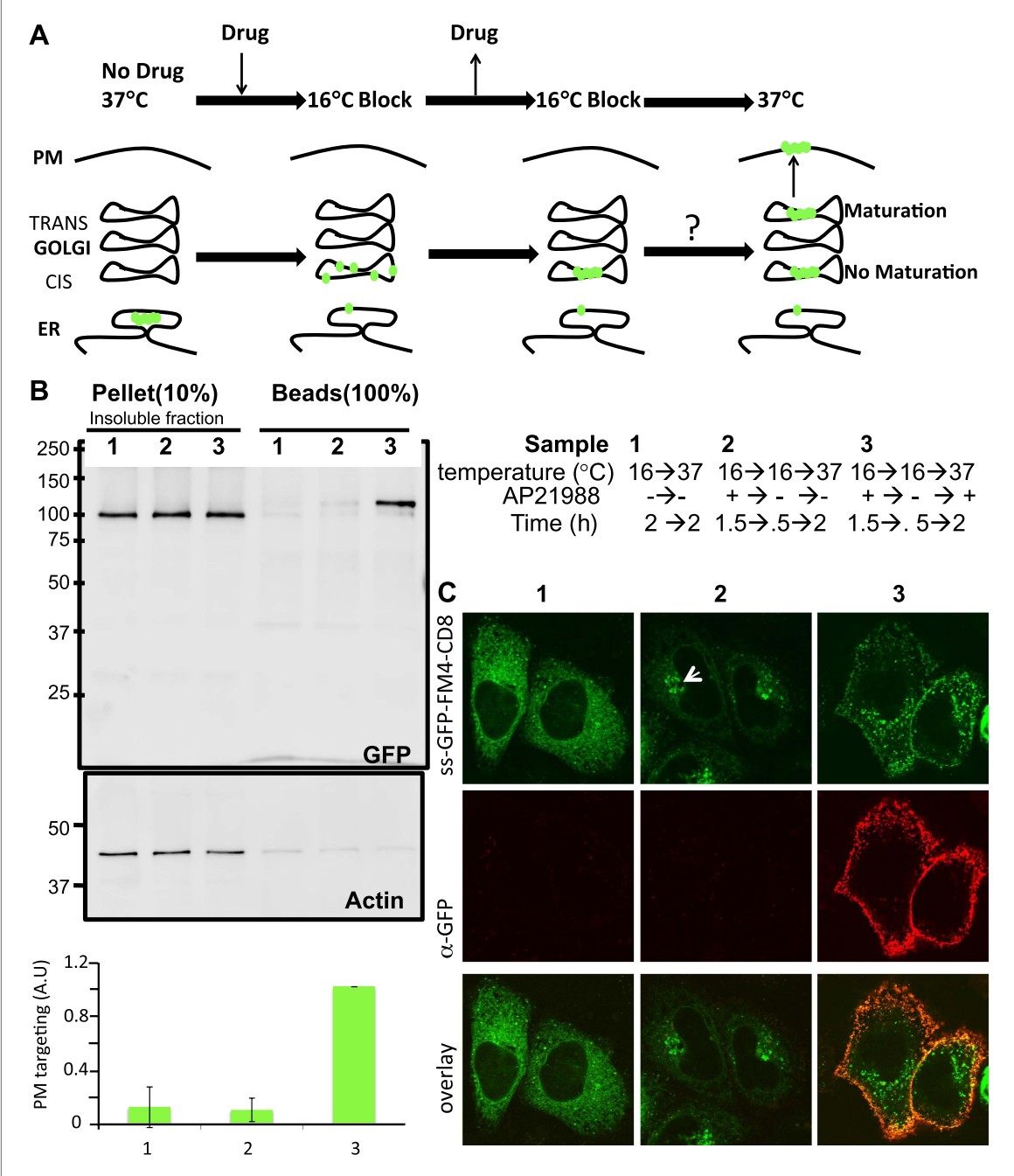

**Figure 3**. Re-aggregated CD8$_{lumenal}$ as a Golgi staple, and its retention within the Golgi. (**A**) General procedure: 16°C temperature block and disaggregation/re-aggregation cycles are combined to form the staples in the cis-Golgi and assess the cisternal progression model. HeLa cells expressing CD8$_{lumenal}$ were incubated for 1.5 hr at 16°C in the presence of the disaggregating drug, and then the drug was removed for 30 min at 16°C to allow re-aggregation in the cis-Golgi. Temperature was shifted to 37°C for 2 hr in the absence (2) or in the presence (3) of the disaggregating drug. As a negative control, cells were incubated at the same temperatures but without any drug at any time (1). (**B**) Immunoblot, after surface biotinylation, showing that disaggregated CD8$_{lumenal}$ (3) is targeted to the plasma membrane (PM), whereas ER-aggregated-CD8$_{lumenal}$ (1), and Golgi-re-aggregated-CD8$_{lumenal}$ (2), and endogenous actin (lower panel) are not. Graph, normalized quantification of PM targeted CD8$_{lumenal}$, with (3) set to 1. Data represent the mean of three independent experiments, (**C**) Confocal micrograph confirming the ER localization of aggregated CD8$_{lumenal}$, (ER staples, 1), the PM localization of disaggregated CD8$_{lumenal}$ (3), and the Golgi retention of re-aggregated CD8$_{lumenal}$ (Golgi staples, white arrow, 2). All experiments were conducted in the presence of cycloheximide.

The following figure supplements are available for figure 3:

**Figure supplement 1**. Glycosylation profile of CD8$_{lumenal}$.

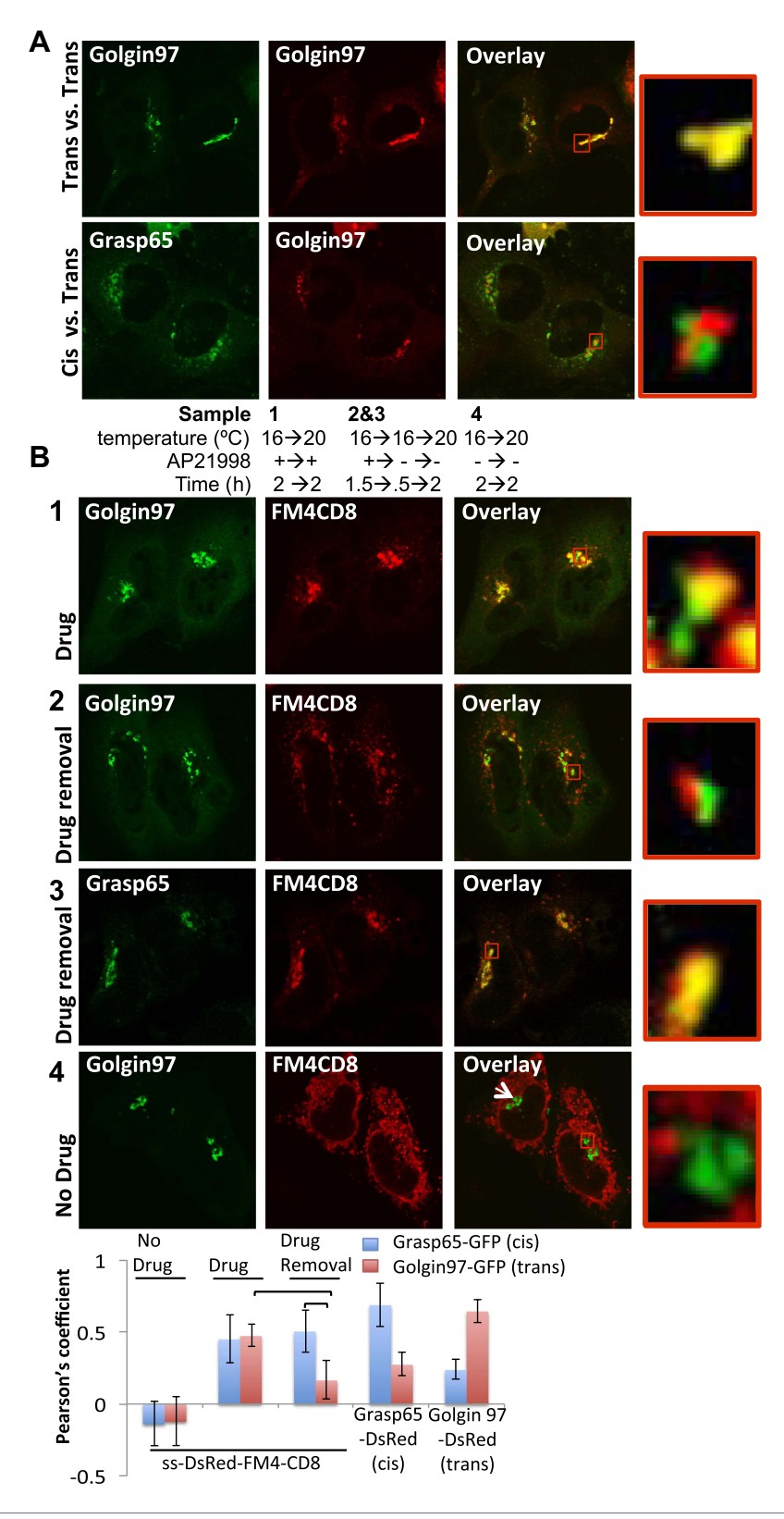

**Figure 4**. cis-Golgi retention of Golgi staples at the light microscopy level. (**A**) Confocal micrograph showing HeLa cells co-expressing DsRed-Golgin97, a trans-Golgi marker, with GFP-Golgin97 (upper panel) or Grasp65-GFP, a cis-Golgi marker (lower panel). Cis- and trans-Golgi are readily distinguishable at the confocal microscopy

*Figure 4. Continued on next page*

*Figure 4. Continued*
resolution. Red square, higher magnification. (**B**) HeLa cells co-expressing DsRed-CD8$_{lumenal}$ with GFP-Golgin97 (1, 2, 4) or Grasp65-GFP (3) were incubated at 16°C for 2 hr with the disaggregating drug, prior to shifting the temperature to 20°C for two additional hours in the presence (1) or in the absence (2, 3) of the disaggregating drug. As a negative control, cells were incubated at the same temperature but without any drug at any time (4). Confocal micrograph showed that Golgi-re-aggregated-CD8$_{lumenal}$, presumably forming cis-Golgi lumenal staples, showed a better co-localization with Grasp65 (3) than with Golgin97 (2). Disaggregated-CD8$_{lumenal}$ showed a stronger co-localization with Golgin97 than with Grasp65 (1), whereas ER-aggregated-CD8$_{lumenal}$ is segregated from both Golgi markers (4). Graph, Pearson's coefficient on the entire field of view, illustrating the co-localization between the different markers. The data represent the mean of three experiments in which a total of >20 fields were counted.

established that this face is the cis-Golgi since the gold particles recognizing the GFP harbored by the staples co-localized with the gold particles that bind GM130, a cis-Golgi marker (*Figure 5E*). Confirming this, the staples were at the opposed end of the stack from clathrin, which is a trans-Golgi marker (*Figure 5F*).

Consistently, lectin-precipitation experiment showed that staples formed at 16°C and chased at 20°C harbored a cis-glycan signature (precipitated by Helix pomatis) but not the trans-glycan signature (not precipitated by Jacalin) (*Figure 3—figure supplement 1*, lane 2)

Altogether, the above experiments reveal that membrane staples formed in the first cisterna of the Golgi stack do not move across the stack, even after several hours.

Interestingly, the staples accumulate at the central portion of the cis-most cisternae (*Figure 5A–C* and *Figure 5—figure supplement 1*) being excluded from their rims. This could be the consequence of the fixed dimension of the staples, which are too small to fit across a dilated rim (*Figure 5D*). So, the staples would be expected to preferentially form in the flattened central regions of a cisterna.

## Membrane staples are static at every level of the Golgi stack

Because large soluble aggregates such as collagens can move forward rapidly across the stack (*Bonfanti et al., 1998*) it was unexpected to observe that membrane staples remain fixed in cis-most cisternae. We therefore wanted to see if staples deposited in later Golgi cisternae might also be static. To deposit staples throughout the Golgi stack we allowed CD8$_{lumenal}$ to enter the Golgi at 20°C in the presence of drug to maintain it in the disaggregated state. Then, the drug was removed at 20°C. Surface biotinylation established that staples formed at 20°C remained at the Golgi even after the cells were warmed up to 37°C for 2 hr (*Figure 5G*), similar to the cis-Golgi staples. Electron microscopy revealed that the staples formed at 20°C were present and retained throughout the entire Golgi stack, including the trans-most Golgi cisterna (*Figure 5H*). This is not surprising because significant back-up of cargo in the stack at 20°C has been previously documented (*van Deurs et al., 1988*).

The fact that the staples are retained in place at every level of the Golgi stack further emphasizes the static nature of the cisterna on the time scale being studied (5–6 hr) and also rules out the caveat that the apparently stable cis localization of staples at 16°C results from anterograde transport followed by efficient retrieval to the cis cisternae.

## Golgi stacks harboring membrane staples are functional

To rigorously conclude that the staples are immobile during ongoing anterograde transport, we need to demonstrate positively that anterograde transport continues in the same stacks that retain the staples. To test this, we introduced a well-studied anterograde cargo into cells harboring staples, the VSV-encoded G protein. Specifically, we used a GFP-tagged version of a temperature-sensitive mutant G protein that is retained in the ER at 39°C, but which can exit the ER and be transported to the cell surface via the Golgi when the temperature is subsequently lowered to 32°C (*Presley et al., 1997*). In these experiments, the staples were labeled with DsRed (DsRed-CD8$_{luminal}$) and co-expressed with GFP-tagged VSV-G. We estimated by confocal microscopy that >95% of the cells were co-transfected. We used the 39°C temperature block to retain VSV-G-GFP in the ER, then a fraction of both VSV-G-GFP and disaggregated DsRed-CD8$_{luminal}$ (+drug) were partially released into the Golgi at 16°C, prior to re-aggregating CD8$_{lumenal}$ (−drug) at 16°C into cis-Golgi staples. Only then was the temperature raised to 32°C for 2 hr to test the transport of VSV-G-GFP (the fraction localized into the stapled cis-Golgi

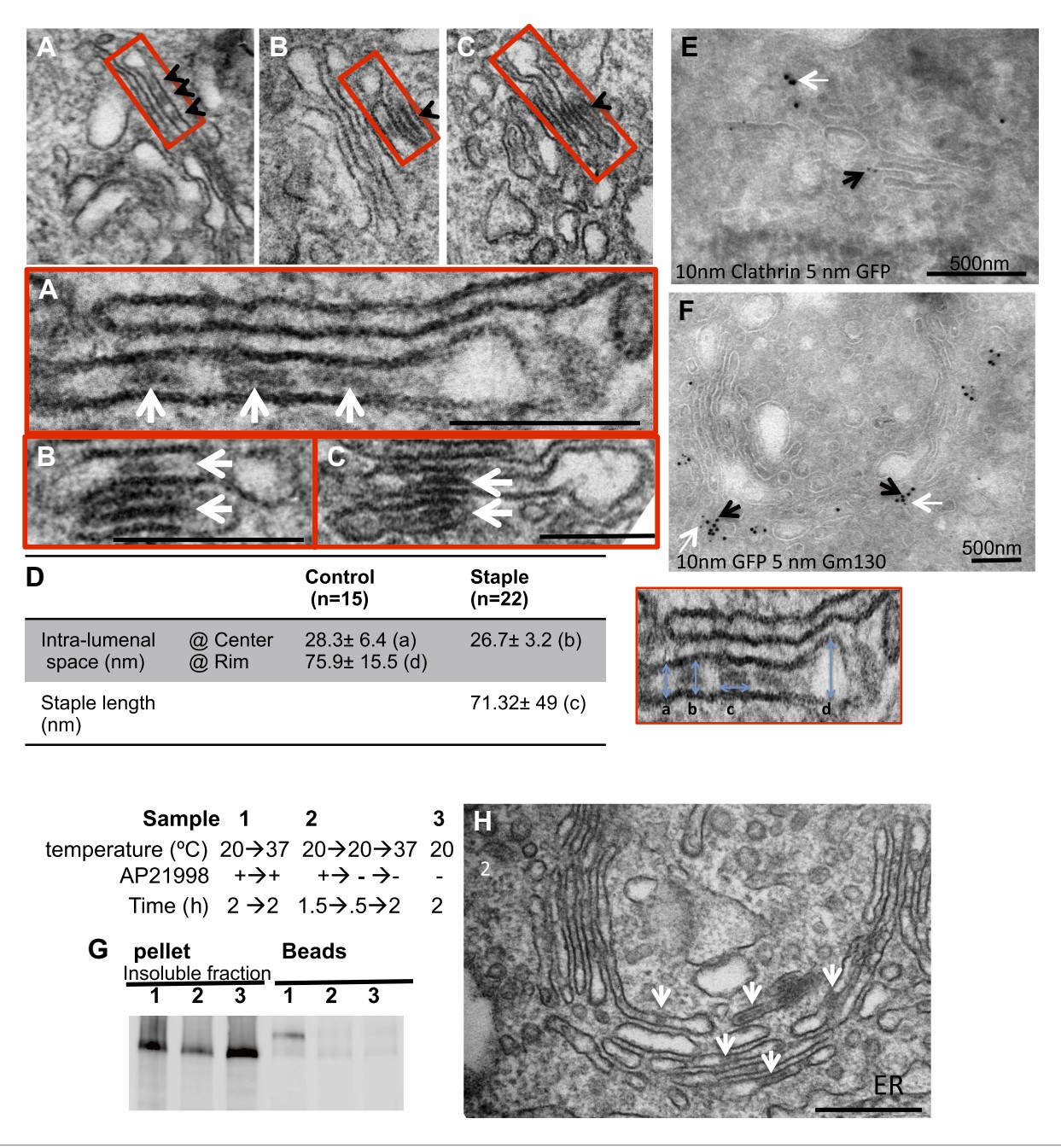

Figure 5. cis-Golgi retention of Golgi staples at the EM level. HeLa cells expressing CD8$_{lumenal}$ were incubated at 16°C in the presence of the disaggregating drug for 2 hr, and then the drug was removed for 30 min, prior to shifting the temperature to 20°C for two to six additional hours. (A)–(C) Electron-micrograph showing that the staples (white arrows) remain at one face of the Golgi, within the first cistern (A) or in the first two cisternae (B and C) after 2 or 6 hr chase at 20°C, respectively. Red square, higher magnification. Scale bar, 200 nm. (D) Intra-lumenal space of stapled Golgi cisternae. Staples that form flat features are localized at the center of the cisternae. Their topological arrangements prevent them from localizing at the rim of the cisternae. (E) Electron-micrograph showing clathrin (10 nm gold particle), which label the trans-Golgi, and the staples (5 nm gold particles) are at two opposite faces of the Golgi. (F) Gm130 (5 nm gold particles), a cis Golgi marker, is at the same face as the staples (10 nm gold particles). (G and H) HeLa cells expressing CD8$_{lumenal}$ were incubated at 20°C (instead of 16°C) in the presence of the disaggregating drug for 2 hr, and then the drug was removed for 30 min, prior to being incubated at 20°C for two additional hours. (G) Immunoblot, after surface biotinylation, showing that the staples formed in the medial/trans Golgi do not reach the PM. (H) Electron micrograph showing that the staples formed at 20°C are actually retained throughout the Golgi and not only at the TGN face.

The following figure supplements are available for figure 5:

Figure supplement 1. Staples within the Golgi.

and the fraction that was still into the ER). Surface biotinylation showed that the amount of VSV-G detected at the surface of cells that harbored staples in their Golgi was 91 ± 12% of the amount of VSV-G detected at the surface of cells harboring a normal Golgi (*Figure 6A*). This showed that the stapling procedure did not interfere dramatically with the transport of classical anterograde cargo.

To be certain that the very Golgi harboring static staples continued to engage in productive anterograde transport, we measured the rate of efflux of VSV-G from Golgi areas harboring staples as compared to those that did not. We used photo-activatable GFP-tagged VSV-G (PAGFP-VSV-G [*Patterson and Lippincott-Schwartz, 2002*]) to activate a cohort of protein within a Golgi area to enable us to measure the release of VSV-G from that same Golgi complex. The rate of the release (half-time ~ 5 min) of PA-GFP-VSV-G from even long-term (4 hr) stapled Golgi was indistinguishable from that measured from a non-stapled Golgi identified with a red-tagged Golgi marker (ST-RFP) (*Figure 6B*). This observation and the fact that stapled-Golgi remain properly stacked established that stapled-Golgi is fully functional for transport.

Then we asked if Golgi-staples could interfere with the transport of endogenous large cargo. We used Saos-2 cells, which secrete endogenously produced collagen-I. After transfection with GFP-CD8$_{lumenal}$, Saos-2 cells were incubated at 20°C in the presence of the disaggregating drug, followed by drug removal to allow the positioning of the staples within the Golgi (*Figure 6C* upper panel, lane 1,2 show the glycosylated form of CD8$_{lumenal}$). 2 hr after stapling the Golgi membranes, the culture media was removed and cells were incubated at 37°C in the presence of fresh media containing ascorbate to release collagen-I from the ER. After a 2 hr chase the amount of newly secreted collagen-I was determined by TCA precipitation/immunoblotting procedure (*Figure 6C*). Cells harboring stapled-Golgi for 2 hr showed 92.5 ± 13% of freshly secreted collagen-I when compared to cells harboring normal Golgi (set to 100%). This suggests that, as reported above with for VSV-G, anterograde transport of Collagen-I is not altered when staples are positioned within the Golgi. To rigorously assess if Golgi staples had only minor effect on secretion, we further tested the rate of transport of collagen-I and MMP2, two proteins secreted by Saos-2 cells. The kinetics of secretion for both proteins did not show any robust differences whether the Golgi apparatus was stapled or not (*Figure 6—figure supplement 1*). Again, this suggests that the amount of staples that is loaded in the Golgi membranes during the re-aggregation procedure does not poison the anterograde transport function of the Golgi.

To further test if Golgi staples interfere with the behavior of Golgi membranes, we assessed the redistribution of Golgi resident enzymes into the ER under BFA treatment (*Lippincott-Schwartz et al., 1989*). Redistribution of Golgi enzymes into the ER was observed whether Golgi staples were present or not (*Figure 6—figure supplement 2*). Note that consistently with the moderate disaggregation reported above (44% solubilization in Triton X100), some aggregates remained in the ER (arrows *Figure 6—figure supplement 2*). These ER-remaining aggregates did not perturb the behavior of the Golgi resident enzymes (Golgi targeting and BFA-redistribution).

Importantly, FRAP experiments (*Figure 6—figure supplement 3*) showed that staples (localized either within the Golgi or within the ER) readily diffuse within the cisternal membranes and do not perturb the diffusion of Golgi resident enzymes; in fact, Golgi-staples diffused only slightly more slowly than disaggregated CD8$_{lumenal}$ at 20°C (*Figure 6—figure supplement 3*). This would explain in a satisfying manner why the essential functions of the Golgi (and ER) such as transport and glycosylation are not perturbed by the presence of the staples, since these behave like any cargo or resident protein within the membrane, as further evidenced by the re-distribution of staples (like any other Golgi cargo or resident enzyme) to the ER as a result of BFA treatment.

Retrograde transport is also a major aspect of Golgi function, and it would potentially be of interest to investigate if staples interfere with such function. While we cannot presently rule out this caveat, even if retrograde transport were blocked it would not diminish our conclusion that anterograde transport of both small and large cargo (next section) can occur at normal speeds without movement of cisternae in the stack.

## Soluble aggregates are transported through stacks in which staples remain

Previously, we found that soluble aggregates that are compositionally similar to the staples (the main difference being that they were not membrane-anchored) rapidly move forward through the Golgi, analogous to collagens (*Volchuk et al., 2000*). We noted that these soluble aggregates were

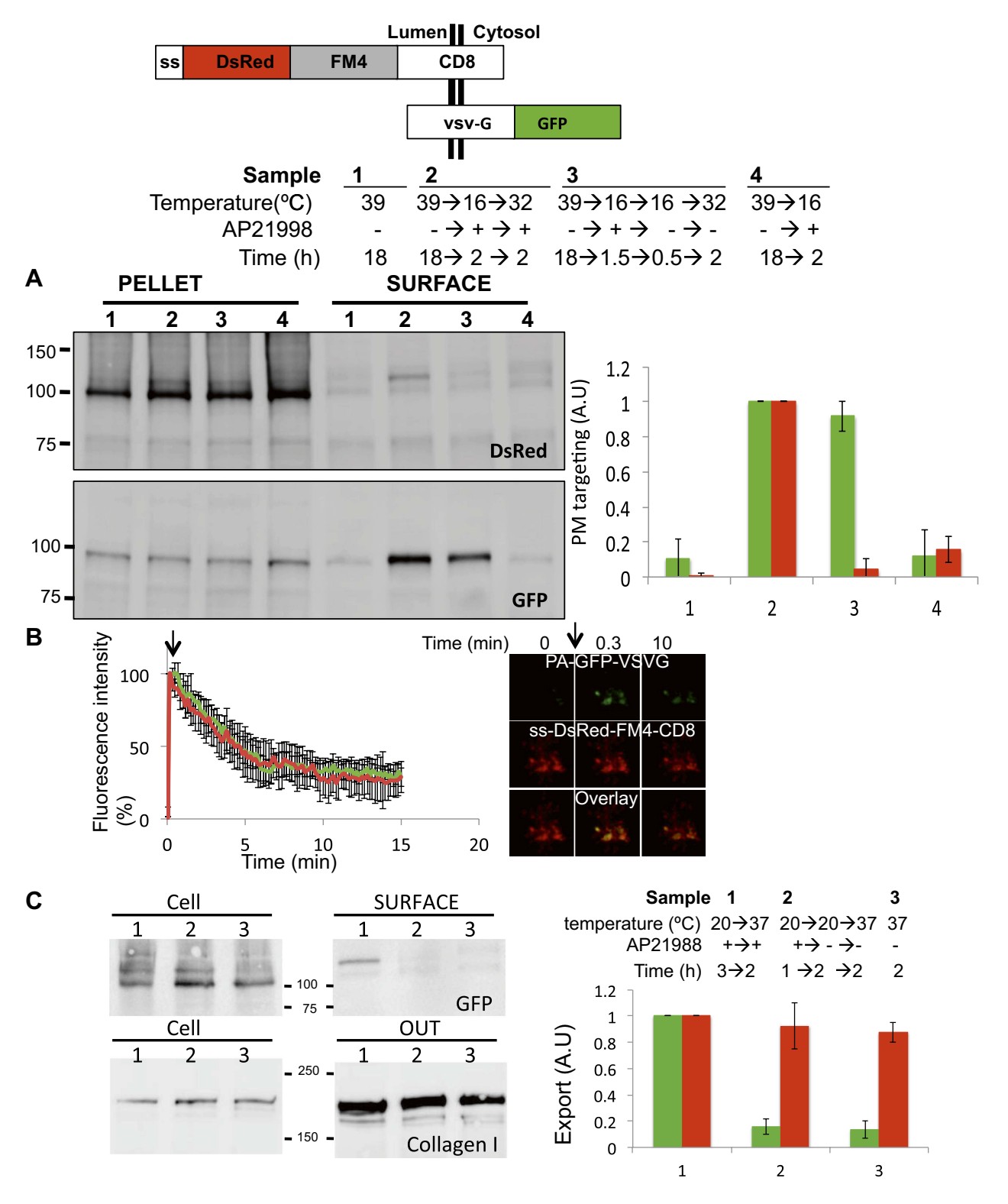

**Figure 6**. Unaltered anterograde transport within short and long-term stapled-Golgi. HeLa cells co-expressing DsRed-CD8$_{lumenal}$ with VSVG-GFP were incubated overnight at 39°C in the absence of the disaggregating drug (1) prior to shifting the temperature to 16°C for 2 hr in the presence of the disaggregating drug (4). Then the temperature was shifted to 32°C in the presence (2) or in the absence (3) of the drug. (**A**) Immunoblot, after surface biotinylation, showing that VSVG-GFP is targeted to the PM regardless of the re-aggregation status of CD8$_{lumenal}$ within the Golgi. Graph, normalized

*Figure 6. Continued on next page*

*Figure 6. Continued*

quantification of VSVG-GFP (green bar) and DsRed-CD8$_{lumenal}$ (red bar). Data represent the mean of three independent experiments. (**B**) HeLa cells co-expressing VSVG-PAGFP and DsRed-CD8$_{lumenal}$ were incubated at 20°C in the presence of the disaggregating drug for 1 hr, prior to removing the drug for 4 hr to form staples throughout the Golgi. VSVG-PAGFP was photo-activated (arrow) within the stapled Golgi and VSVG-PAGFP fluorescent signal was measured over time at 32°C. HeLa cells co-expressing VSVG-PAGFP with ST-RFP were used as a control. Confocal micrographs illustrating the release of VSVG-PAGFP from long-term stapled-Golgi. Graph, normalized fluorescence quantification of VSVG-PAGFP over the time in stapled Golgi (green) and control Golgi (red). Data represent the mean of at least three different experiments. (**C**) Saos-2 cells expressing GFP-CD8$_{lumenal}$ were incubated at 20°C in the presence (1, 2) or in the absence (3) of the disaggregating drug. When required Golgi staples were formed for 2 hr at 20°C (2), prior to shifting the temperature to 37°C in the presence of ascorbate to allow collagen-I release. Fresh media was added at the beginning of the chase. Immunoblot, after surface biotinylation and TCA precipitation of the media, showing that collagen-I is secreted whether or not staples are pre-positioned in the Golgi (2) or ER (3). Cell fractions that are used as loading control correspond to the Triton X-100 soluble fraction after sonication (10% input lysate). Graph, normalized export of CD8$_{lumenal}$ (green bar) and Collagen-I (red bar). Data represent the mean of two independent experiments.

The following figure supplements are available for figure 6:

**Figure supplement 1**. Golgi staples do not inhibit the rate of secretion of Collagen-I and MMP2.

**Figure supplement 2**. Golgi staples do not prevent redistribution of Golgi membranes within ER mediated by BFA.

**Figure supplement 3**. Staples are laterally mobile and do not perturb lateral diffusion of Golgi resident enzymes.

concentrated at the rims of the Golgi cisternae, analogous to collagens. Given the prima facia contradiction between our current observation of immobile staples and our prior observation of mobile soluble aggregates composed of virtually the same protein, we repeated the earlier studies and compared the fates of soluble aggregates and staples in the same cells.

We first asked whether lumenally-expressed soluble FM4 (FM4-hGH) and membrane-bound FM4 proteins (CD8$_{lumenal}$) co-aggregate in the cell. This would be expected because they share the same homotypic adhesion FM modules. As expected, when the two FM-containing proteins were present in the ER they were efficiently co-immunoprecipitated from cell extracts (***Figure 7—figure supplement 1***). However, when these same proteins were disaggregated and allowed to leave the ER and then re-aggregated for 1 hr in the cis-Golgi at 16°C, or even further in the Golgi stack at 20°C, the Golgi-modified forms of these same proteins were no longer co-immunoprecipitated (***Figure 7—figure supplement 1***). On the contrary homo-aggregates constituted of soluble cargo (hGH) were efficiently formed even after re-aggregation at 16°C or 20°C (***Figure 7—figure supplement 1***). This suggests that hetero-aggregates are not formed efficiently within the Golgi cisternae.

Confocal microscopy independently established that the soluble and membrane-bound FM proteins segregated into separate aggregates within the same Golgi area (***Figure 7—figure supplement 2***) and, electron microscopy confirmed that this separation occurs even within the same cisternal compartment. Even when both reside in the same cisterna, the electron-dense plaques comprising the staples are localized in the central regions of the cisterna while the soluble aggregates are concentrated at the dilated rims (***Figure 7E***).

What are the fates of soluble aggregates as compared to staples in the Golgi when anterograde transport is permitted to resume at 37°C? Whereas the Golgi retained the staple protein it released the soluble aggregates to the secretory pathway (***Figure 7A***). This biochemical result confirms yet again that hetero-aggregates (composed of staples and soluble FM proteins) were not formed efficiently within the Golgi, confirming by yet another approach that the two FM proteins aggregate separately within the Golgi cisternae (even when using a 2:1 membrane:soluble protein ratio).

Electron microscopy was also used to follow the fate of the two types of aggregates. When aggregates accumulated in the cis-Golgi are allowed to move anterograde through but not exit the Golgi following warm-up to 20°C, ≈75% of the staples, as expected, remain at the cis-face of the Golgi, whereas ≈75% of the soluble aggregates are transported to the opposite, trans-face of the same Golgi stacks (***Figure 7B*** and graph).

We conclude that the anterograde transport process permits large soluble aggregates to be carried forward between cisternae that are immobile.

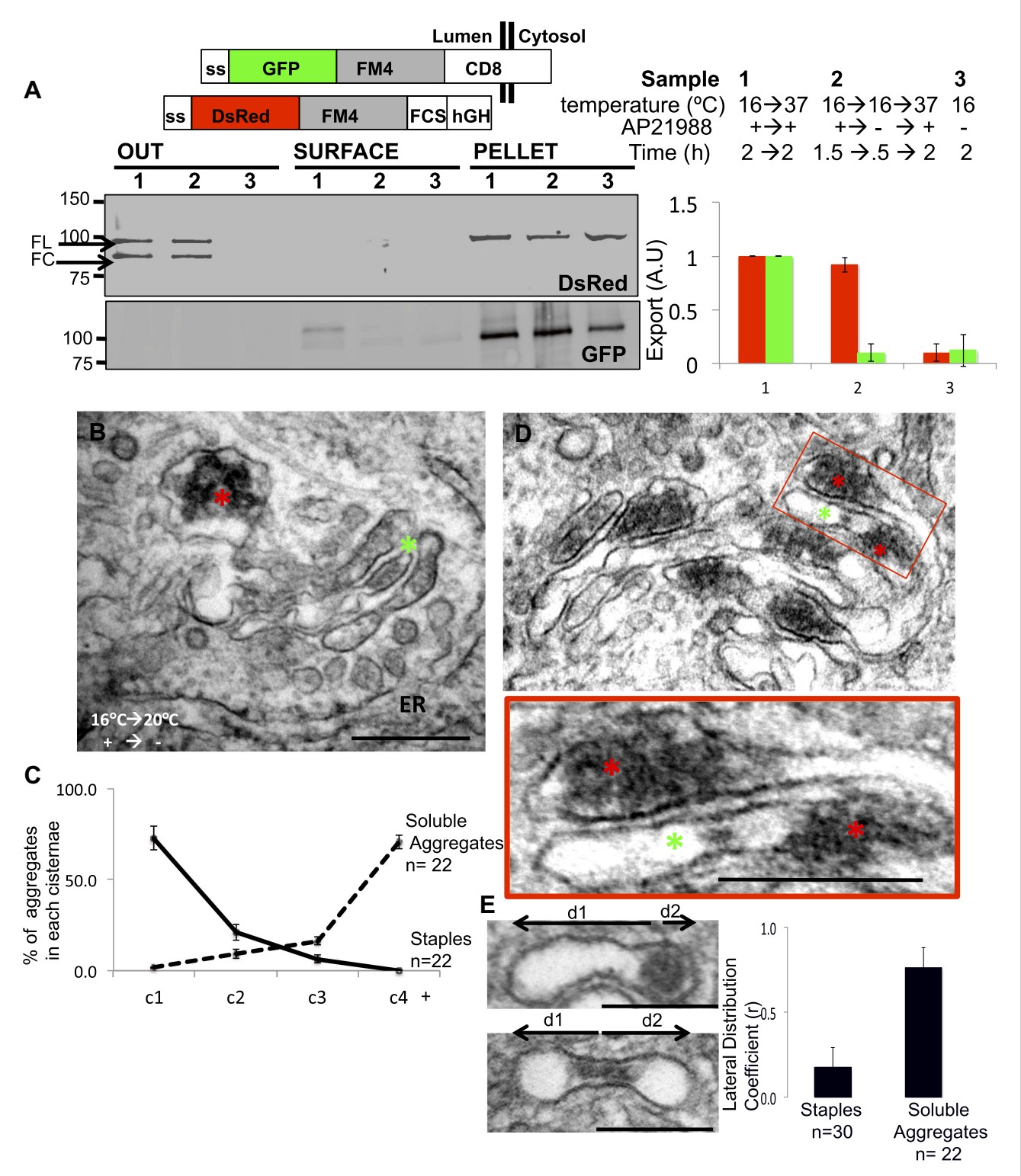

**Figure 7**. Unaltered transport of soluble aggregates within stapled Golgi. HeLa cells co-expressing GFP-CD8$_{lumenal}$ and DsRed-hGH, both harboring the FM-aggregation domains, were incubated at 16°C for 2 hr in the presence of the disaggregating drug (3, negative control) prior to shifting the temperature to 37°C in the presence (1) or in the absence of the drug (2). (**A**) Immunoblot, after surface biotinylation and TCA precipitation of the media, showing that DsRed-hGH is secreted regardless of its own re-aggregation status, or the re-aggregation status of CD8$_{lumenal}$. Prior to being secreted disaggregated and re-aggregated DsRed-hGH are cleaved by furin. FL, full-length isoform of DsRed-hGH. FC, furin-cleaved isoform of DsRed-hGH. Graph, normalized quantification of PM targeted CD8$_{lumenal}$ (green bar) and normalized secreted DsRed-hGH (red bar). Data represent the mean of three

*Figure 7. Continued on next page*

*Figure 7. Continued*

independent experiments. (**B**) Cells were incubated at 16°C for 2 hr in the presence of the disaggregating drug, then the temperature was shifted to 20°C for two additional hours in the absence of the drug. Electron micrograph showing Golgi staples (green star) that remains at the cis-Golgi, which is adjacent to ER membranes, whereas the soluble aggregates (red star) is localized to the opposite face of the Golgi. (**C**) Graph, distribution of staples and soluble aggregates within the Golgi stack: cells expressing CD8$_{lumenal}$ or hGH were subjected to the re-aggregation procedure at 16°C, prior to being incubated at 20°C for 2 hr and prepared for classical EM. The distribution of the aggregates (soluble and staples) within the Golgi stack is shown as a percentage of the total in Golgi areas. (**D**) Electron micrograph showing segregation between Golgi lumenal staple (green star) and soluble aggregates (red star). Golgi staples form one flat feature in the center of one cisterna whereas the soluble aggregates form spherical aggregates that are localized at the rims of the cisternae throughout the Golgi. (**E**) HeLa cells expressing hGH or CD8$_{lumenal}$ were subjected to the re-aggregation procedure at 16°C for 2 hr prior to being processed for conventional EM. The upper panel shows the distribution of the soluble aggregates at the rim of the cistern, the lower panel illustrates the central positioning of the staple. Graph, lateral distribution coefficient (r) of staples and soluble aggregates within Golgi cisternae. The distances that separate the middle of the aggregates from each end of the cisterna (d1 and d2) were measured, and (r) was calculated as follow $r = (d1 − d2)/(d1 + d2)$ with $d1 \geq d2$. As a result, (r) → 0 for object at the center, whereas (r) → 1 for object at the rim.

The following figure supplements are available for figure 7:

**Figure supplement 1**. Co-immunoprecipitation of ER-staples and ER-soluble aggregates.

**Figure supplement 2**. Segregation of staples from soluble aggregates within the Golgi at the light level.

## Discussion

Does the presence of staples perturb anterograde transport? On the one hand, because staples freely diffuse like any other membrane constituent it is hard to imagine how a moderate amount of staples would broadly perturb Golgi transport or function such as glycosylation, and there is no positive evidence from our studies that they do so. On the other hand, because they span and link between two sides of the cisterna, it is easy to see why staples should remain in the thin centers of the cisternae, far from the dilated rims (whose width is too great to accommodate the staples) where transport vesicles bud, and also why staples would be sterically excluded from budding COPI or other vesicles if they did occasionally reach a rim, and thus would remain within the cisterna.

With this in mind, our findings can be simply summarized as follows: 1) membrane staples localize to the central regions of Golgi cisternae where they segregate from large soluble aggregates which mainly concentrate at the dilated rims; and 2) membrane staples remain in place as anterograde transport occurs, whereas large soluble aggregates and small cargo are transported at normal rates.

The obvious explanation of the above is that while the cisternae in the stack (containing staples mainly in their central regions) are static, the rims of cisternae—mainly containing the soluble aggregates—are mobile (at least on the average) in the anterograde direction, a process we term 'rim progression'.

However, the unique nature of the artificial staples prevents us from definitively concluding that 'pure maturation' does not occur in region of the Golgi stacks that are depleted of staples. Although we established by using broad sets of assays that anterograde transport through stapled Golgi is not altered, we cannot rule out that staples introduced, non-obvious, minor side effects on Golgi.

It would be very easy to mis-interpret rim progression as cisternal progression if the only marker being studied is in the rim (***Bonfanti et al., 1998***). In retrospect, the principal evidence for cisternal progression in Golgi stacks was the mobility of large aggregates like collagens that are located to the rims (***Bonfanti et al., 1998***). But in retrospect a stricter interpretation of that data would have been to conclude that the 'at least some portions' of the cisterna containing the collagen aggregates (i.e., the rims) are mobile, without reaching any conclusion with respect to the rest of the cisterna. A striking feature of the collagen data is that while the majority of the aggregates are indeed rapidly transported, a sizable minority (~20% [***Bonfanti et al., 1998***]) remains in the Golgi cisternae, mainly in the centers of the cisternae, consistent with our results and interpretation.

We do not know how the rims of otherwise stable stacks progress, but the most likely possibility would be cycles of fission and fusion that are known to take place in the Golgi (***Pfeffer, 2010***). Pfeffer has considered how this could be organized in the cis → trans direction by known Rab cascades in the Cisternal Progenitor model, which could well be the basis for the transport of rims dilated with aggregates. We would imagine that after fission a transiently separated rim would nonetheless remain in

register in the stack because its membrane would still be engaged in stacking/tethering interactions. Pulse-chase experiments (*Volchuk et al., 2000*) showed the appearance of detached rims closely-associated with the edges of the stack that transiently appeared during the transport of soluble aggregates (which we termed mega-vesicles at the time). Retention of rims in transit would facilitate orderly rim progression.

To the extent that the bulk of a Golgi stack is static on the time scale of anterograde transport, this reduces the quantitative demand on retrograde transport of Golgi resident proteins and the need for cisternal maturation. In budding yeast, where cisternae generally do not stack, the process of cisternal maturation is well documented, though formal proof that anterograde cargo is present in maturing cisternae is still lacking (*Losev et al., 2006*; *Matsuura-Tokita et al., 2006*). We do not wish to conclude that cisternal maturation does not occur in the stacked Golgi of complex plants and animals, or that it may not be important in re-assembly of the Golgi after cell division, only that it does not appear to be required on the time scale of biosynthetic protein transport in interphase in stacked Golgi.

How are smaller cargoes that do not create dilated rims transported across the Golgi if the cisternae are static? We propose that rim progression is dedicated to large soluble aggregates that due to physical constraints are forced into the rims away from adhesive centers of cisternae, and differs mechanistically from vesicle budding. COPI vesicles are known to bud extensively from the rims of Golgi cisternae at every level and contain–in addition to retrograde-directed cargo–anterograde-directed cargo such as insulin, albumin, and VSV-G (*Rothman, 2010*); so this biochemically well-understood pathway (*Popoff et al., 2011*) would seem to be the principal contender. There may well be other, parallel pathways depending on physiological state, specifically budding/fusing tubules, which have been observed especially under conditions of protein over-expression (*Ladinsky et al., 1999*; *Trucco et al., 2004*).

Our findings, along with other evidence (*Brugger et al., 2000*; *Patterson et al., 2008*), point to a broad division of the Golgi cisternae into two functionally distinct domains: the centers and the rims. The rims specialize in exit and entry into the stack and are dynamic, while the centers are static and adherent to form the stack and specialize in biosynthesis and processing. Much remains to be learned about how these domains segregate and organize the multitude of tasks associated with Golgi function.

## Materials and methods

### Cell culture and transfection

HeLa cells were maintained at 37°C in 5% $CO_2$ in DMEM (Gibco, Grand Island, NY, United States) supplemented with 10% FBS (Gibco). Saos-2 cells were maintained with the same media supplemented with ascorbate (25 µg/ml). HeLa cells were transfected using lipofectamine 2000 (Invitrogen, Grand Island, NY, United States) as recommended by the manufacturer. Saos-2 cells were transfected using electroporation (Nepa21type II model from Nepa gene, Chiba, Japan).

### Plasmids

$CD8_{lumenal}$ (ss-GFP-FM4-CD8 and ss-DsRed-FM4-CD8) were generated by sequential insertion of GFP (or DsRed), and CD8 encoding sequences into a pC4 ss-FM4 backbone vector (ARIAD), using XbaI/SpeI compatibility and BamH1 restriction site. CD8 was amplified from pC4-CD8-GFP plasmid (*Lavieu et al., 2010*). A similar strategy was used to create ss-DsRed-FM4hGH and ss-GFP-FM4hGH. PAGFP-VSVG and $_{ts}$VSVG-GFP were ordered from Addgene. J Rohrer provided ST-RFP. Grasp65 and Golgin97 encoding plasmids were a gift from Y Wang and S Munro, respectively.

### Cell surface biotinylation, TCA precipitation, lectin precipitation, SDS/PAGE and immunoblot analysis

These experiments were performed as described previously (*Lavieu et al., 2010*) with slight modifications. HeLa cells were grown in six-well plates. All the disaggregation/re-aggregation experiments were performed in HBSS (Gibco) supplemented with 10% FBS (Gibco) and 100 µg/ml cycloheximide (Sigma, St Louis, MO, United States). Disaggregating drug (AP21998; ARIAD, Cambridge, MA, United States) was used at 1 µM. 16°C, 20°C, 32°C and 37°C incubations were performed using temperature-controlled incubators.

Briefly, subconfluent HeLa cells were biotinylated in 1 mg/ml EZ-Link Sulfo-NHS-SS Biotin at 4°C for 30 min and washed two times with PBS. After neutralization with 100 mM glycine pH7 at 4°C for 30 min, cells were washed, detached and collected by centrifugation. Pellets were resuspended in lysis buffer (1% Triton X-100, 5 mM EDTA, 150 mM NaCl pH7.4 and 1% protease inhibitor) for 1 hr at 4°C

and sonicated (80W, 10 s pulse). The cell lysate was centrifuged, and the supernatant was collected. Samples were incubated with neutravidin agarose resin at 4°C overnight for Biotin-labeled protein precipitation. The neutravidin interacting proteins were eluted with SDS sample buffer from neutravidin pellets. Extracted proteins were first separated in SDS-polyacrylamide gels and then transferred onto nitrocellulose membranes for immunoblotting. After blocking with fat-free milk, the membranes were incubated with appropriate primary antibodies. Primary antibodies were detected by chemiluminescence using horseradish peroxidase-conjugated secondary antibodies. Fluorographs were quantitatively scanned using the NIH image software.

When TCA precipitation was required, 0.1% FBS was used. Chase media was collected and precipitated overnight at 4°C with 10% TCA. After centrifugation TCA pellets were washed with acetone prior to being resuspended in loading buffer and analyzed by immunoblotting blot as described above.

For lectin precipitation, HeLa cells extracts containing $CD8_{lumenal}$ were incubated at 4°C with Jacalin-immobilized lectin (Sigma-Aldrich) or Helix pomatia Gel-HPA-immobilized lectin (EY Laboratories, San Mateo, CA, United States) prior to being processed for immunoblotting as described above. For immunoblotting, we used monoclonal anti-GFP (Roche, Brandford, CT, United States), polyclonal anti-mCherry (Biovision, Milpitas, CA, United States), anti-Collagen-I (SP1.D8 from Developmental Studies Hybridoma Bank, Iowa City, IA, United States), and polyclonal anti-MMP2 (Cell Signaling, Boston, MA, United States).

## Confocal imaging

HeLa cells were grown on glass coverslip in 24-well plates. For surface labeling, cells were incubated on ice for an hour with a monoclonal anti-GFP antibody (Roche) prior to being fixed for 10 min with 4% PFA and incubated with an Alexa Fluor 546-coupled antibody. Cells were then processed for confocal microscopy imaging, which was performed in multi-tracking mode on either a Zeiss LSM510 or a Zeiss LSM510 META. Images were analyzed using Zeiss LSM510 software or using ImageJ (co-localization finder plugin). For photo-activation experiments we used a multicolor spinning-disk confocal based on an inverted Olympus microscope (IX-71) and Perkin-Elmer Ultraview system with 5-laser and FRAP/photoactivation. VSVG-PAGFP was photo-activated within the Golgi region (identified by the co-expressed DsRed-tagged protein) at 405 nm (80% laser power) for 5 s. VSVG-PA-GFP was imaged using 488 nm light (10% laser power).

## Electron microscopy

### Sample fixation for regular epon embedding

The cells were fixed in 2.5% gluteraldehyde in 0.1 M sodium cacodylate buffer pH7.4 for 1 hr at room temperature. They were rinsed in sodium cacodylate, scraped and pelleted in 2% agar, then post-fixed in 1% osmium tetroxide for 1 hr, en bloc stained in 2% uranyl acetate in maleate buffer pH5.2 for a further hour, rinsed then dehydrated in an ethanol series and infiltrated with resin (Embed812 Electron Microscopy Science) and baked over night at 60°C. Hardened blocked were cut using a Leica UltraCut UC7, 60 nm sections were collected onto formvar/carbon coated nickel grids and stained using 2% uranyl acetate and lead citrate.

### Sample fixation for immunoelectron microscopy

Samples were fixed in 4% paraformaldehyde/0.1% gluteraldehyde in phosphate buffered saline for 15 min then post-fixed with just 4% PFA in PBS for 1 hr. Cells were scraped and re-suspended in 10% gelatin, then placed in cryoprotectant 2.3 M sucrose overnight at 4°C. These were transferred to aluminum pins and rapidly frozen in liquid nitrogen. The frozen block was trimmed on a Leica Cryo-EMUC6UltraCut and 65 nm thin sections were collected. The frozen sections were thawed and collected on grids and floated in PBS ready for immunolabeling.

### Immunolabeling of sections

The grids were placed section side down on drops of 0.1 M ammonium chloride for 10 min to quench untreated aldehyde groups, then blocked for nonspecific binding on 1% fish skin gelatin in PBS for 20 min.

Single labeled grids were incubated on either a primary antibody rabbit anti-Gm130 (Abcam) 1:10, rabbit anti-Clathrin (Millipore) 1:100, mouse anti-GFP (Clontech) 1:50, or rabbit anti-GFP (Invitrogen) 1:100 dilutions for 30 min. Grids were rinsed and placed on protein A gold (PAG) 10 nm (UtrechtUMC) for 30 min. Double labeled grids were incubated with mouse anti-GFP antibody with 5 nm PAG and then the rabbit anti-Clathrin with 10 nm PAG, or rabbit anti-GFP with 10nmPAG and mouse

anti-Gm130 with 5 nm PAG. All grids were rinsed in PBS, fixed using 1% gluteraldehyde, then rinsed and transferred to a UA/methylcellulose drop for 10 min. Samples were all viewed on FEI Tencai Biotwin TEM at 80Kv. Images were taken using Morada CCD and iTEM (Olympus) software.

## Acknowledgements

We thank ARIAD Pharmaceuticals for providing us with its secretion/aggregation kit reagents. Thanks to Derek Toomre for providing access to the spinning disk microscope and the electroporation apparel. Thanks to Daniel Kuemmel for modeling the trans-complexes. Thanks to Morven Graham and Xinran Liu for providing EM assistance. Thanks to Pietro De Camilli for the gift of the Clathrin antibody. Thanks to Myun Hwa Dunlop for editing the manuscript.

## Additional information

### Funding

No external funding was received

### Author contributions

GL, Conception and design, Acquisition of data, Analysis and interpretation of data, Drafting or revising the article, Contributed unpublished essential data or reagents; HZ, Acquisition of data, Contributed unpublished essential data or reagents; JER, Conception and design, Analysis and interpretation of data, Drafting or revising the article

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
