## [Decision Letter]

Thank you for sending your work entitled “Dynamics of the Golgi Apparatus I: Stapled Golgi Cisternae Remain in Place as Cargo Passes Through the Stack” for consideration at *eLife*. Your article has been favorably evaluated by a Senior editor and 3 reviewers, one of whom is a member of our Board of Reviewing Editors, who believe that with appropriate revision the work should be published by *eLife*. The comments relate to the experiments studying collagen release in stapled cells. Specifically:

1) There is no known cargo of this type and that has to be clearly stated. If “stapling” results in a “dead” Golgi then there might be a transient block in traffic at the time of drug addition (“stapling”) until a new region of Golgi had formed. Please state the half life of the collagen compared with the staples in stapled cells to try to address this point.

2) Some staples may be in cis and others trans. This should be mentioned in the text.

3) In Figure 6, it was not clear that the collagen secretion was due to new collagen versus previously secreted collagen. This must be clarified or carried out in such a way that only newly secreted collagen is scored.

4) Figure 6 and related text: the experiments assume the cells are co-transfected. This is important for interpreting the results, and the authors have assumed cells tend to express both VSV-G and CD8 constructs. The authors should measure and include the degree of co-transfection.

5) Figure 7 and related text: how much FM protein was actually expressed? A Western blot is needed to show this. A plasmid ratio is given but this isn't particularly helpful in determining the actual protein ratio of membrane:soluble FM protein.

---

## [Author Response]

As requested prior to peer review, we initially assessed if Golgi staples altered the secretion of collagen. We now clarify the minor concerns that were raised during peer review.

*1) There is no known cargo of this type and that has to be clearly stated. If “stapling” results in a “dead” Golgi then there might be a transient block in traffic at the time of drug addition (“stapling”) until a new region of Golgi had formed. Please state the half life of the collagen compared with the staples in stapled cells to try to address this point*.

We have shown that the rate of secretion of collagen-I and MMP2 are unchanged whether or not staples are pre-positioned in the Golgi (Figure 6—figure supplement 1). Since there is no retention of collagen-I, there is no reason to expect any change in the lifetime of this protein. On a more technical point of view, how can one measure the intracellular lifetime of a protein, which is efficiently secreted?

We now state in the revised manuscript that the unique nature of the artificial staples prevents us from definitively concluding that “pure maturation” does not occur in regions of the Golgi stacks that are depleted of staples. Although we established by using broad sets of assays that anterograde transport through stapled Golgi is not altered, we cannot rule out that staples introduced other, non-obvious, minor side effects on Golgi membranes.

*2) Some staples may be in cis and others trans. This should be mentioned in the text*.

Do you mean cis-Golgi vs Trans-Golgi localization or cis-interaction vs trans-interaction?

Concerning the localization within the Golgi we combined biochemistry (lectin precipitation), confocal microscopy, electron microscopy, and immuno-electron microcopy to establish that staples formed in the cis-Golgi remain in the cis Golgi.

Concerning the mode of interaction, although we visualized only staples resulting from trans-interaction, as proved by the artificial stacking of ER membranes harboring cytosolic staples (Figure 1—figure supplement 1), we cannot rule out that staples may form in cis (on the same membrane bilayer) when physical constraints are favorable. This is now mentioned in the manuscript.

*3) In Figure 6, it was not clear that the collagen secretion was due to new collagen versus previously secreted collagen. This must be clarified or carried out in such a way that only newly secreted collagen is scored*.

We scored only the secretion of newly released collagen-I. In Figure 6, staples were formed for two hours and then secretion of collagen-I was tested. At the beginning of the chase (t=0), old media was removed and replaced with fresh media. After two hours the media was collected and submitted to TCA precipitation. This is now clarified in the manuscript.

Note that Figure 6—figure supplement 1 showed the rate of release of collagen-I (and MMP2), and clearly showed the absence of collagen-I in the media at t=0.

*4) Figure 6 and related text: the experiments assume the cells are co-transfected. This is important for interpreting the results, and the authors have assumed cells tend to express both VSV-G and CD8 constructs. The authors should measure and include the degree of co-transfection*.

When observed under a confocal microscope, >95% of the cells are co-transfected. This is now reported in the manuscript.

*5) Figure 7 and related text: how much FM protein was actually expressed? A Western blot is needed to show this. A plasmid ratio is given but this isn't particularly helpful in determining the actual protein ratio of membrane:soluble FM protein*.

We estimated that a 3:1 membrane:soluble plasmid ratio results in a 2:1 membrane:soluble protein ratio.

Such quantification seems trivial; however, the molecular weight of ssDsRedFM4hGH (sol) and ssDsRedFM4cDB (rMD) are almost similar, making it difficult to distinguish with the same antibody each protein when they are co-expressed (even when using an acrylamide gradient gel: Figure 8).Author response image 1

Therefore we could only estimate by densitometry the sum of the chemiluminescence signal emanating from both proteins. The amount of plasmid used for the soluble protein was constant, whereas the amount of plasmid for the membrane protein was increased by a factor 3. Intensity of membrane protein expressed alone was set to 1. Using this method we showed that using a 1:1 TMD:Sol plasmid ratio increases the signal by a factor 1.9 (in theory a 2 fold increase is expected). When the TMD:Sol plasmid ratio is 3:1, the signal increased by a factor 2.8, suggesting a 1.8:1 TMD:Sol protein ratio. This ≅2 protein ratio is now reported in the revised manuscript.